# Community-Based Interventions in People with Palliative Care Needs: An Integrative Review of Studies from 2017 to 2022

**DOI:** 10.3390/healthcare12151477

**Published:** 2024-07-25

**Authors:** Antonia Vélez-López, Juan Manuel Carmona-Torres, Ángel López-González, José Alberto Laredo-Aguilera, David Callado-Pérez, Joseba Rabanales-Sotos

**Affiliations:** 1Primary Healthcare Local Office in Isso, 02420 Albacete, Spain; antonia.velez@alu.uclm.es; 2Escuela Internacional de Doctorado, University of Castilla-La Mancha, C/ Altagracia, 50, 13071 Ciudad Real, Spain; 3Department of Nursing, Physiotherapy and Occupational Therapy, Faculty of Physiotherapy and Nursing, University of Castilla-La Mancha, Av de Carlos III s/n, 45004 Toledo, Spain; josealberto.laredo@uclm.es; 4Multidisciplinary Research Group in Care (IMCU), University of Castilla-La Mancha, Campus de Fábrica de Armas, Av de Carlos III s/n, 45004 Toledo, Spain; 5Department of Nursing, Physiotherapy and Occupational Therapy, Facultad de Enfermería, University of Castilla-La Mancha, Campus Universitario s/n, 02071 Albacete, Spain; angel.lopez@uclm.es (Á.L.-G.); joseba.rabanales@uclm.es (J.R.-S.); 6Group of Preventive Activities in the University of Health Sciences (GAP-CS), University of Castilla-La Mancha, University Campus s/n, 02071 Albacete, Spain; 7Complex Chronic Patient Service at Sant Joan de Déu Hospital, 08950 Barcelona, Spain; dcallado@sescam.jccm.es

**Keywords:** community-based intervention, community participation, home care services, nursing, palliative care, quality of life, Spain

## Abstract

Aim: To describe the latest scientific evidence regarding community-based interventions performed on patients in need of palliative care worldwide. Introduction and background: Given the rise of chronic diseases, their complexities and the fragility of patients, we are facing around 56.8 million people in need of palliative care. Community-based healthcare, particularly palliative care, can address social inequalities and improve the biopsychosocial health of disadvantaged populations. Therefore, primary care, as the main health referent in the community, has a central role in the care of these patients. Methods: This is an integrative review from January 2017 to June 2022 that follows the PRISMA statement and has been registered in PROSPERO. PubMed, Cuiden, the Web of Science (WoS), Cochrane and LILACS were the five databases searched. The scientific quality assessment of the articles was carried out following the CASPe methodology. Study selection was carried out by two researchers, A.V.L. and J.M.C.T., using the inclusion and exclusion criteria mentioned below. In cases of doubt or discrepancy, a third author (J.R.S.) was consulted. Results: The interventions mentioned in the 16 articles analysed were classified under the following categories: music therapy, laughter therapy, spiritual and cognitive interventions, aromatherapy, interdisciplinary and community-based teams, advance care planning and community, volunteering, telemedicine and care mapping. Example: Educating people to talk about different ethical issues could improve their quality of life and help develop more compassionate cities. Conclusions: We have identified interventions that are easily accessible (laughter therapy, telemedicine or music therapy), simple enough to be carried out at the community level and do not incur high costs. This is why they are recommended for people with palliative care needs in order to improve their quality of life.

## 1. Introduction

Palliative care is the kind of care that improves the quality of life of patients and their families in a holistic way who are facing problems inherent in incurable diseases. It prevents and relieves suffering through early identification, correct assessment and treatment of pain and other problems, whether physical, psychosocial or spiritual [1]. According to the World Health Organization (WHO), there are about 56.8 million people in need of palliative care, but only 14% of them actually receive it [1]. Palliative care needs will continue to rise due to the increase in chronic diseases, an ageing population and new treatments; and also, because these scientific advancements extend life expectancy and, in turn, chronicity. Integrating a sustainable, quality and accessible palliative care system requires working with primary healthcare, community and home care providers, as well as supporting care providers such as family members and community volunteers [1]. Primary care, as the main health referent in the community, has a central role in the care of these patients [2].

Community-based care reduces the number of hospitalisations, increases the quality of care and is satisfying for patients [3]. Moreover, the latest trends focus on preventing complications and promoting people’s autonomy [4,5]. Unfortunately, this type of care is very heterogeneous, as it depends on the zone where one lives, which creates inequalities in access to health resources and increases isolation. These issues can be overcome thanks to community-based care teams, care networks and community resources [3,6]. Community-based healthcare, particularly palliative care, can address social inequalities and improve the biopsychosocial health of disadvantaged populations [6,7].

By means of a new approach towards care, education, training and policy formulation, the community can be empowered through the development of volunteering and service networks that can help these people with palliative care needs [8]. Being close to the community allows primary care and palliative home care teams to detect barriers or needs in the community, and then take steps or implement programmes that can improve the quality of life of people [9].

Article 17 of Law 4/2017, of March 9, on the rights and guarantees of people in the dying process, mentions the importance of providing spiritual and emotional care in the hospital and community environment to caregivers, family members and patients alike. Health centres and institutions will facilitate, at the request of patients, or their representatives or relatives, access to people who can provide spiritual support, highlighting home care that provides comfort to people with palliative care needs [8].

In general, these people live in a community and wish to be at home with their family and next to their neighbours [10]. At times, the family or main caregivers are not capable of giving their relatives adequate support, so they come under physical and emotional strain [11]. Previous studies demonstrate the importance of the support of primary care teams [3,10,12] who are capable of accompanying patients, training and empowering their families by providing resources about care management, and supporting the biopsychosocial needs of the community [13].

In some cities in Spain, there are associations and organizations that carry out interventions in the community, but these are not related to end-of-life or palliative care. This is why we need to identify the existing resources in our area, as well as the different types of community-based interventions (music therapy, laughter therapy, volunteering, etc.) for people with palliative care needs, so we can address those and support them and their families throughout their illness [2].

Community-based palliative care faces several significant challenges. First, resource constraints are a major obstacle. A lack of adequate funding and shortages of essential medical supplies hinder the delivery of quality palliative care. In many areas, interdisciplinary teams lack the necessary equipment to manage complex symptoms, affecting patients’ comfort and quality of life [14,15,16].

Second, geographic disparities complicate access to palliative services. Rural and remote areas often have fewer specialized care providers, forcing patients and their families to travel long distances to receive care, if it is available at all. This can be especially difficult for patients with limited mobility or those in advanced stages of their disease [14,15,16].

Third, staffing shortages are a critical concern. There is a lack of trained palliative care professionals, which increases the workload for the few who are available; this can lead to professional burnout. This situation also limits the ability to provide personalized and appropriate care to all patients in need [14,15,16]. These challenges highlight the need for research and policies that address these gaps, improve funding, and encourage the training and equitable distribution of the palliative health workforce.

As mentioned above, the global prevalence of palliative care needs is increasing, and the World Health Organization (WHO) estimates that, annually, more than 56.8 million people need palliative care, including 25.7 million people in the last year of life [1]. As is well known, providing interventions to people in need of palliative care improves the quality of life for individuals and families. It is necessary to know what palliative care interventions are being developed in the community, as this would help to optimize the resources available in society and provide solid evidence to guide policy and clinical practice. In fact, few studies have been carried out in this area.

The aim of the present study was to identify community-based interventions on patients in need of palliative care in order to improve palliative care in the community and make use of the resources available around a town in the Albacete region (Spain).

## 2. Methods

### 2.1. Design and Sources of Information

An integrative review benefits a systematic review of community-based interventions in palliative care by combining quantitative and qualitative evidence, providing a more comprehensive view of the topic. This allows us not only to assess the effectiveness of interventions, but also to understand the experiences and perceptions of patients and caregivers. Integrating multiple data sources and methods improves our understanding of the contexts and factors that influence the success of interventions, which can better guide clinical practice and policy formulation.

To carry out this study, an integrative review of a descriptive nature was conducted on the scientific evidence from January 2017 to June 2022, following the Preferred Reporting Items for Systematic Reviews and Meta-Analyses statement (Appendix A) [17,18]. The review has been registered in the PROSPERO platform with the ID number CRD42023418065.

There were five databases searched during this review: PubMed, Cuiden, the Web of Science (WoS), Cochrane and LILACS (the most important international databases in nursing).

### 2.2. Search Strategy

To do the searches, the research question was formulated following the PIO format (population, intervention, outcomes). Those searches were carried out between October 2022 and June 2023, in order to incorporate the latest interventions.

The clinical question was which community or training interventions (I) can help improve palliative care (in terms of quality of life) (O) for patients with palliative care needs in the community or at home (P).

A community health intervention is a planned, collaborative strategy that aims to improve the health of individuals within a specific community. These interventions address health problems through active community participation, multisectoral collaboration and the implementation of programmes and activities tailored to the needs and characteristics of the local population.

The same search string was used in the different databases searched. The search string included the Boolean operator AND and the DeCS (Health Science Descriptors, for its Spanish acronym) terms (Table 1).

### 2.3. Inclusion and Exclusion Criteria

The following inclusion criteria were considered:Scientific papers published in the last 5 years, to focus on the latest evidence.Papers written in Spanish and English.Papers that include interventions applicable to patients in need of palliative care and at a community level.Papers directed towards both the paediatric and adult populations.Qualitative, descriptive and interventional studies, systematic and integrative reviews, meta-analyses and clinical cases.

The exclusion criteria were as follows:Research papers not applicable at a community level.

### 2.4. Study Selection

Study selection was carried out by two researchers, A.V.L. and J.M.C.T., using the inclusion and exclusion criteria mentioned above. In cases of doubt or discrepancy, a third author (J.R.S.) was consulted. The initial results of the literature search included 4652 articles from the LILACS, PubMed, Cuiden, Cochrane and WoS databases (Figure 1). After a first screening by title and abstract, 92 articles were left. After a full-text review and the quality assessment with the CASPe [19] tool, 16 articles were selected for data collection and analysis of results.

### 2.5. Studies Quality Assessment: Detection of Possible Biases

In order to assess the methodological quality of the selected documents, the CASPe [19] checklists were used, and their questions were adapted to each study design type. The CASPe checklists do not assign a numerical value to each study, but they help determine the validity and reliability of its design and analyse the strengths and weaknesses of clinical trials. None of the eligible studies were excluded from the review due to quality concerns. All included studies were assessed for quality and independently reviewed by two reviewers (AVL and JRS. Reliability between reviewers was high, and all discrepancies were discussed until an agreement was reached. As a measurement of methodological quality, it was determined that each study should have a 50% positive rating.

### 2.6. Results Extraction

For the data collection from the selected articles, an ad hoc table was created, containing the following information: title, authors and year of publication, study type, intervention type, sample size, results, conclusions and study quality (Table 2).

### 2.7. Data Analysis

The data were analysed to compare outcomes between different community interventions that can improve the quality of life in patients with palliative needs at a community level.

### 2.8. Ethics Approval and Consent to Participate

All methods were carried out in accordance with relevant guidelines and regulations. No institutional or other licensing committee’s approval was needed for guideline creation, as participants were not subjected to procedures and were not required to follow rules of behaviour.

## 3. Results

A total of 16 articles were selected. These included one longitudinal study, one cohort study, one cross-sectional study, five systematic reviews, five qualitative studies, two mixed-methods studies and one quasi-experiment study.

The selected studies included a total population of 11,517 palliative care patients. The studies were carried out between 2018 and 2022, in the following locations: Brazil [21,22], Spain [10], Belgium [12,23], Argentina [24], Korea [20], China [11], Portugal [25], Singapore [26], Australia [27] and France [3].

### 3.1. Thematic Synthesis

The unit of analysis comprises 16 articles that focus on different interventions that can be made in the community to improve the quality of life of people with palliative care needs and their families.

#### 3.1.1. Music Therapy

In their article, Sousa, Silva & Paiva highlight the importance of music therapy, physical exercise, massage, the use of therapeutic toys and early nursing consultation for the treatment of pain, anxiety and fatigue. However, there are some negative results due to the nursing staff’s lack of training, technical skills and emotional expertise for the application of these activities in patients with palliative care needs [21]. Music therapy can naturally alleviate pain, provide relief and comfort, and improve the relationship between patients with palliative care needs and their families through the expression of emotions. It is one of the non-invasive interventions with minimum side effects that crosses cultures and binds them together [28].

#### 3.1.2. Laughter Therapy

Interventions such as laughter therapy at home, performed by clowns, have been shown to lower anxiety and sadness levels, improving the quality of life and the domain of social activities thanks to social support. It has been noted that it is important to finance these kinds of interventions and to carry out further research to help develop these activities and raise awareness of them [19]. The use of humour as a therapy improves communication and connection among patients, their families and the professionals who face end-of-life experiences every day. It encourages the ability to escape the issues and burdens derived from the illness. Furthermore, they emphasise the importance of using humour as a means of communication and expression, which helps social interactions. There are not enough studies about this topic [29].

#### 3.1.3. Cognitive Interventions

Cognitive–existential therapy for caregivers encourages self-care, reduces compassion fatigue and improves the caring experience of people in need of palliative care. Rational emotive therapy and logotherapy promote social skills and human strengths. The aim is to detect different everyday issues and situations and to teach how to overcome them.

In the study by Hidalgo-Andrade et al., an intervention consisting of eight sessions was conducted using Viktor Frankl’s logotherapy and Albert Ellis’ rational emotive therapy as theoretical frameworks. The intervention was developed in response to specific needs identified in a previous qualitative study on compassion fatigue and the job satisfaction of formal caregivers. These interventions improve the quality of life of caregivers, which in turn results in a better quality of care and support for people in need of palliative care [20]. The intervention has been shown to be effective in the long term in reducing compassion fatigue, making it replicable and promising. It is suggested that future studies evaluate its effectiveness in settings outside of palliative care to extend its applicability.

#### 3.1.4. Aromatherapy

One study about aromatherapy, reflexology and massage concludes that even if these therapies are highly valued by patients, there is not enough scientific evidence to prove their efficacy. The problem might be the study’s approach, since the researchers do not reflect on any negative or dangerous aspects of these interventions [30].

#### 3.1.5. Interdisciplinary Community-Based Teams

Interventions related to the identification of patients with palliative care needs and interdisciplinary home care by primary and secondary care teams improve cost management (programme sustainability and avoiding hospitalisations) and the care experience for providers, patients and their families [10]. These teams, made up of professionals from various disciplines, perform comprehensive assessments, coordinate care, educate patients and caregivers and provide regular follow-ups. Among the services offered are medical and nursing care, rehabilitation therapies, palliative care, nutritional support and social services. They assist with daily activities such as bathing and grooming and provide supervision to ensure patient safety.

When home and community care interventions are implemented after team building and the integration of shared medical records, and are widely disseminated and promoted, they can improve resource management and quality of life for the patient’s family, while ensuring that the patient’s wish to die at home and in the community is respected [12]. In Korea, most cancer patients expect to receive home care. Palliative care teams provided palliative care training and increased awareness of palliative care, and also demonstrated relief of pain, anxiety and depression in these patients [20]. In France, it was concluded that home-based interventions help reduce the number of hospitalisations, which increases the quality of care and patient satisfaction. Community-based interventions include external resources, such as infrastructure, and internal resources related to healthcare professionals, such as time management and the use of the home as a source of information about the patient’s condition [12].

#### 3.1.6. Advance Care Planning

Advance care planning interventions alleviate pain, as well as the economic and psychological burden of caregivers and family members. In order to provide this kind of care, training and education are needed to avoid fear in the communication between providers and their patients and relatives. Nurses have a privileged position in this therapeutic relationship since they are the closest to the patients and they spend the most time caring for them. It is important to highlight the refusal to implement this measure due to a lack of human resources and Chinese culture and legislation [11]. It is necessary to eliminate the prohibition to ask people in need of palliative care how they wish to die; they need to be able to make autonomous decisions and talk about them with their families so that an excellent level of care is ensured while respecting their wishes. It is necessary to organize nation-wide campaigns to promote talking about death and respecting decisions [26]. There were positive aspects detected in interventions such as education on symptoms for patients with chronic obstructive pulmonary disease, advance care planning, a holistic approach to illness, breathlessness management, etc. However, studies were inconclusive and further research with enough power is needed to confirm that these people benefit with a higher quality of life [31].

### 3.2. Community, Volunteering, Telemedicine and Care Mapping

Educational intervention programmes implemented in a teenage population can be a great resource to raise awareness about palliative care, as well as to encourage and engage community participation in such types of care. Educating people to talk about different ethical issues could improve their quality of life and help develop more compassionate cities [23]. By determining who provides palliative care, how and where they do it and its related problems, communication, early identification of illnesses that require palliative care and education can all be improved. Palliative care provision maps can improve the lives of neighbours from a certain area who are in need of care [23]. Volunteers carry out supporting interventions in the community as a supplement to professional caregiving. They need adequate training and education since they will be on their own at the home of a person who is at the end-of-life stage or who has already done advance care planning [27].

The most significant findings regarding the types of intervention are the following: music therapy can relieve pain, provide relief and improve relationships through the expression of emotions. Laughter therapy uses humour as a means of communication and expression, which promotes social interactions. Cognitive interventions can detect problems and teach coping skills. Aromatherapy may reduce anxiety and pain, although studies are inconclusive. Interdisciplinary home care teams improve the care experience by respecting the will of patients to die at home while reducing healthcare costs. Advance care planning could reduce pain and economic and psychological burdens by avoiding the fear of communication and improving social relationships. Regarding volunteering, telemedicine, community and care maps can be a great resource for raising awareness and promoting quality palliative care, while engaging citizens to participate in the community.

## 4. Discussion

This integrative review confirms that implementing community-based interventions for people in need of palliative care and their families can improve their quality of life and provide relief at home throughout the illness process.

Interventions that can be implemented at a community level, such as music therapy [21,28] or laughter therapy [20,29], provided there has been previous training on their application, can encourage emotional expression and foster distraction or even relaxation, which are needed to face the suffering of having an incurable disease. The development of clown activity projects provided a significant improvement on the level of social support and social relationships in general, because smiling helps one regain one’s confidence [22].

Spiritual and cognitive interventions [24] also bring benefits to people in need of palliative care and their families. The sick patients start expressing suffering or concerns about the purpose of their existence, so these interventions seek anything that can help them find meaning in life, understand the irreversible nature of the process, express emotions, or get protection and relief. We take care not only of the body, but also of the mind. This study was an eight-session psychoeducational intervention that combined existential and cognitive therapy techniques to decrease compassion fatigue and increase compassion satisfaction in formal caregivers. Thanks to advance care planning [11,23,29], we can sense how patients prefer to die, and even where and with whom. Different scenarios can be considered throughout the process of the illness. This is the decision-making process of informed patients who have been trained and guided by their healthcare providers. Being aware of this is essential to respect the people who are at a terminal stage, and to reduce pathological grief or complicated grieving processes in their families or caregivers, by providing them with much-needed confidence towards the end of the process.

Care mapping and volunteering [23,25,27] are community-based interventions currently at their peak, and they allow us to include the community in the caring process. Terms such as patient training, population empowerment, motivation or resource mapping show that it is the people who have the power to care for their health, and that they just need the providers’ support to be able to attain it. Concepts such as compassionate cities advocate such power and stress the importance of identifying the resources available in each neighbourhood, helping improve people’s lives and sharing responsibilities.

Interdisciplinary care [3,12,13,14] and volunteer collaboration [27] enable high-quality home-based care at a lower cost for the healthcare system. The SARS-CoV-2 pandemic stressed the importance of telemedicine, like virtual support or consultation and urgent care in real time. Technology and telemedicine can transform community-based palliative care by improving access to specialized care, regardless of geographic location. They enable remote symptom monitoring, virtual consultations, and ongoing support, reducing the need for hospitalisations and in-person visits. This facilitates more personalized and timely care, improving patients’ quality of life and easing the burden on caregivers. In addition, telemedicine can optimize resource utilization and expand the reach of palliative services to underserved areas. For example, they can help diagnose skin diseases, or through videos, analyse respiration or identify possible dysphagia. There are also teaching resources available such as recordings of parents changing a tracheostomy or instructions on how to programme a feeding pump [32,33,34]. Having an interdisciplinary team that can address all aspects of care of a person who has palliative care needs is fundamental to the improvement of the experience during the illness, both for the patient and their caregivers or loved ones. Support during their life, final moments and at their deathbed is needed.

### 4.1. Study Limitations

We found clear limitations to interventions, such as aromatherapy or reflexology [35]. This is because there are not enough studies that prove the efficacy of these interventions in improving the quality of life or reducing pain in patients, and because of the approach taken in these studies. The heterogeneity in the types of studies included is to be highlighted.

In this integrative review, only two studies carried out in Spain were found. One dealt with the importance of developing cognitive–existential therapies as a means to reduce compassion fatigue in caregivers, while the other studied the impact of integrated palliative care programmes. All of the above makes us reflect on the types of community-based interventions that nurses currently implement in Spain and how interesting it would be to apply interventions that have already been studied in other countries.

The strength of this study is the possibility it provides of integrating these findings into a strategy (provided a previous mapping of palliative care resources and interventions is done in the community). The strategy would include clinical education, active listening and the promotion of health-related activities by primary care offices together with palliative care units, since doing so in the community has low costs. With this strategy, we could avoid issues that affect patients’ families, such as depression, anxiety or a sense of defeat. With respect to the patient, as we have already mentioned, we can improve their quality of life, help them express their emotions, avoid unnecessary hospitalisations, reduce loneliness, etc.

These interventions are not mutually exclusive, and their combination can strengthen their effects in terms of meeting the needs that arise in the community environment. For example, the combination of laughter therapy, aromatherapy and music therapy is accessible, does not incur additional costs and is easy to do in the home setting within the community.

### 4.2. Implications for Nursing and Health Policies

Nurses are the health professionals who make the greatest number of home visits, many of them to patients requiring palliative care. This article has identified interventions that nurses can apply in patients’ homes. There is a need for health policy makers and health managers to implement these interventions in their care programmes, especially in primary care, as they have been shown to be easy and inexpensive to implement. In fact, these interventions would reduce healthcare costs by improving the quality of life of patients and their caregivers, thereby reducing the number of visits needed and the costs they generate.

For example, for a community-based palliative care educational intervention, the target population could be selected in coordination with the local palliative care team. Taking advantage of the proximity and continuity of care of primary care nurses, schools for caregivers and patients can be organized with the aim of improving their experience of care, with organizers collecting suggestions including concerns or areas they want to work on and organizing it in a community way, accessible and adapted to the environment and people’s habits. Activities such as training in home care, stress management, etc. can be programmed.

There are currently few community projects in palliative care. We are going through a process of change in the medical model of care, where community care [29] is gaining more and more prominence, but it is not specialized in palliative care. We still have a long way to go, to eliminate taboos, to bring death and illness closer to the community and to work together so that people may die in the place they wish and with the people they wish to be with.

## 5. Conclusions

Community-based interventions included as part of the care of people with palliative care needs improve the quality of life for both the sick person and their caregivers or family members.

We have detected interventions that can be done in the community and at home, which reduce health costs, improve the overall experience during the illness process, and foster companionship in care (volunteer networks, education about end-of-life care, etc.).

We have also identified interventions that are easily accessible (laughter therapy, telemedicine or music therapy), simple enough to be carried out at home and that do not incur high costs. They could be of great help in communities supported by interdisciplinary palliative care teams who work with associations and organizations from the neighbourhoods together with volunteers (spiritual interventions, advance care planning, etc.).

Community-based interventions can alleviate needs that arise in people with palliative needs, accompanying and providing the best care adapted to the environment and the resources that people in that geographic area have. They can also reduce inequalities in healthcare, because patients are not affected by a geographic gap or distance and can use available community resources and the support of their primary care team together with specific interdisciplinary palliative care teams to enjoy their last moments in the place they want to be and with the people they want to be with. Due to the scarce number of articles about the topic studied, we conclude that further research is necessary to discover more types of community-based interventions and determine their influence on the domain of palliative care.

As future research, we propose studies that analyse the benefits of establishing palliative care training for adults and children in the community setting, promoting compassionate cities as a tool for empowerment in rural settings. Also, it could be interesting to analyse the effect of the implementation of workshops or seminars on palliative care in rural settings as part of university healthcare degrees in order to have professionals in the future who are well trained to meet patient needs in palliative settings and can create high-resolution primary care consultations.

## Figures and Tables

**Figure 1 healthcare-12-01477-f001:**
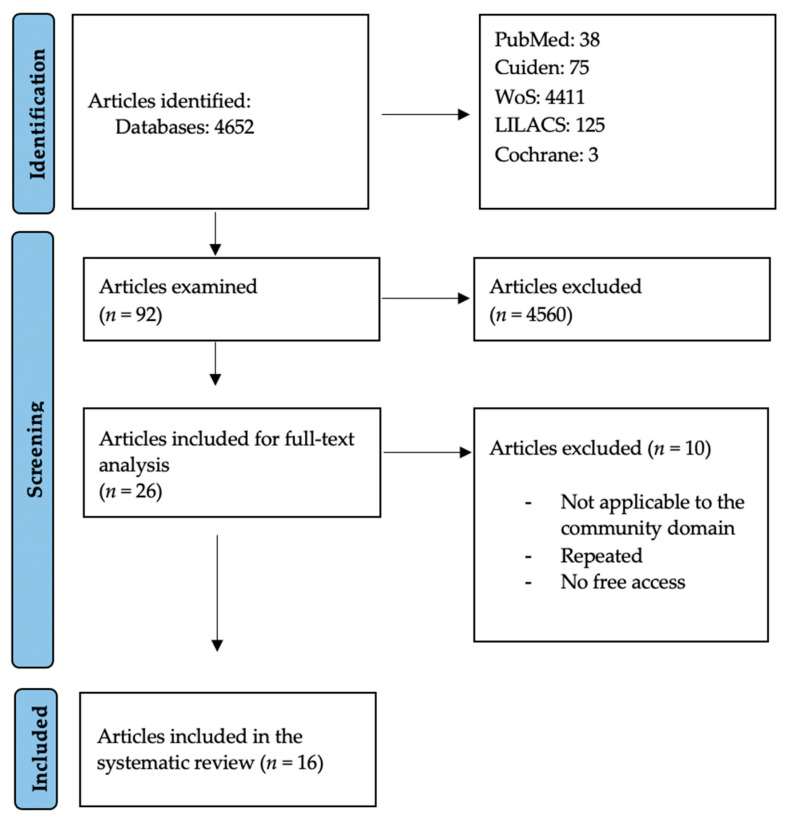
PRISMA flow diagram of literature search procedure (source: PRISMA) [15,16].

**Table 1 healthcare-12-01477-t001:** Search string used in the different databases.

Database	Search String
PubMed Cuiden Web of Science Cochrane LILACS	(“Community participation” OR “training community” “OR “home interventions”) AND “palliative care” “Quality of life AND palliative care” (“leisure activities” OR “alternative therapy” OR “spiritual therapy”) AND “palliative care”

**Table 2 healthcare-12-01477-t002:** Summary of the articles comprised in the unit of analysis.

Authors and Year of Publication	Study Type	Participants	Intervention Types	Results	Conclusions	Study Quality as per CASPe Tool
Boudy (2020) [3]	Qualitative study	21 family doctors	Community home-based care.	The resources used by family doctors: Internal resources: medical experience, skills and education.	The promotion of palliative home care should be a matter of social responsibility, capable of improving the quality of life and the experiences throughout the illness. Strengthening community resources and reducing exhaustion in healthcare providers could be more important than medical knowledge.	8/10 items
				External resources: healthcare services, assistance platforms and workers. Home palliative care can be both a burden and, at the same time, a satisfying and meaningful activity. This balance may be more important than medical education.		
Larrañaga (2019) [10]	Comparative cross-sectional study	1023 (year 2012) and 1142 (year 2015) patients requiring palliative care	Common clinical pathways between primary and secondary care, training courses, patient classification.	The likelihood of being identified as a palliative care user is 2.4 times higher when the patient identification programme is applied, through McNamara’s minimum estimation.	Correct patient identification reduces the number of hospitalisations and improves clinical activity.	9/10 items
Xing (2021) [11]	Phenomenological qualitative study	13 community health services centres	Advance care planning, multicultural care.	Some of the factors that negatively affect the quality of palliative care include Chinese legislation, insufficient allocation of human resources and training, and deep-rooted traditional beliefs about death.	Advance care planning can improve the doctor–patient relationship in terms of communication and quality of care.	8/10 items
Choi (2018) [20]	Longitudinal and cross-sectional comparative study	100 patients: 50 patients in the experimental group who received continuous palliative care and 50 patients in the control group who received outpatient palliative care.	Education on palliative care, home care, promotion of care.	The Community-based Palliative Care Project was composed of a multidisciplinary team and, by establishing community networks, improved the quality of life of patients and raised awareness of such type of care.	Patients who receive palliative care have lower levels of anxiety, a higher quality of life and reduced health costs.	9/10 items
Maetens (2019) [12]	Cohort study	17,674 patients, of whom 11,149 received home palliative care support in the last 720-15 days of life.	Multidisciplinary home care.	56% of people who used home palliative care support died at home, compared to 13.8% of those who did not use home palliative care support. Costs are reduced by avoiding unnecessary diagnostic testing and hospitalisations by about EUR 1617.	The support and care provided by palliative home care teams reduce health costs and improve the quality of life for both patients and their caregivers.	10/11 items
Silva and Sousa (2019) [21]	Integrative review of evidence	18 articles	Music therapy, massage, physical exercise, toys, early nursing intervention to treat a specific symptom, nurse education and training.	It was highlighted that these interventions showed excellent results to treat pain, anxiety and fatigue.	The promotion of nursing training was suggested in order to equip nurses with more skills and provide them with necessary emotional support when they are handling palliative care.	8/10 items
Santos (2021) [22]	Quasi-experiment study	16 patients from two units of the Family Health Strategy, who fulfilled the assigned inclusion criteria	Laughter therapy with clowns.	Improvement in quality of life and social support.	Clown therapy can improve the quality of life and the social support of people who need palliative care at home.	9/10 items
Vanderstichelen (2022) [23]	Qualitative study	254 answers	Trained volunteers, training and education on palliative care.	Eighty per cent of associations have volunteers. The most important difficulty is finding new and competent volunteers. Thirty-three per cent of the associations offer compulsory training on palliative care and nursing.	Organizations are encouraged to invest in training and providing support to volunteers. These actions are carried out with the aim of providing palliative care in the community to those who need it, while also benefiting the volunteer.	10/10 items
Hidalgo-Andrade (2020) [24]	Mixed-methods study	84 formal caregivers	Existential and cognitive therapy.	Rational emotive therapy and logotherapy promote social skills and human strengths.	These interventions improve the quality of life of caregivers, which in turn results in a better quality of care and support for people in need of palliative care.	8/11 items
Martins Pereira (2018) [25]	Mixed-methods study: action research study	69 people from a community parish in Portugal	Awareness about palliative care, education, advancement and encouragement of compassionate communities.	The palliative care awareness programme was well received and left a positive message about community-based palliative care.	Educational intervention contributes to raising awareness about palliative care. Further research and interventions across all age groups are needed in order to promote engagement in care and to empower the population.	9/11 items
Menon (2018) [26]	Qualitative study	61 health providers, caregivers and patients	Establish advance care planning, empower families and people with palliative care needs, adapt advance planning to suit all cultures.	Healthcare providers, caregivers and patients were unaware of advance care planning and the legal framework governing it, but they were amazed about it and were willing to implement it.	Patients and caregivers need to be involved in decision-making and informed about it from the beginning of the illness, in order to improve the therapeutic relationship.	7/10 items
Saurman (2019) [27]	Qualitative study	15 interviews with health professionals	Community-based care, resource allocation and integration.	The importance of establishing networks between services and healthcare providers to improve communication and palliative care has been demonstrated.	Awareness of the network of care services and healthcare providers in the community can help health providers to improve the level of care they provide.	8/10 items
Nyashanu (2021) [28]	Systematic review	8 articles	Music therapy.	Music therapy improves quality of life, spirituality, happiness and hope, and reduces pain, anxiety and depression.	It was concluded that music therapy can be an effective psychosocial approach that positively impacts people’s biopsychosocial wellbeing.	7/10 items
Linge-Dahl (2018) [29]	Systematic review	13 studies	Humour, laughter therapy.	A positive effect of humour on patients, their caregivers and family has been recorded.	An appropriate use of humour can improve the quality of life of patients, their family and caregivers. Further studies to support this are needed.	9/10 items
Candy (2020) [30]	Systematic review	172 articles	Aromatherapy, reflexology and massage therapies.	The use of these interventions to reduce pain and anxiety, and to improve the quality of life, yielded some results, although these were inconclusive. Reduced sample size and number of studies.	The use of reflexology yielded good results.	10/10 items
Broese (2021) [31]	Systematic review	31 articles	Support with chronic obstructive disease and education about it, emotional management.	The patients widely accept these kinds of interventions thanks to the support for the care received. The main obstacle is the timing of referral.	Further high-quality studies are needed in order to assess the importance of palliative interventions in patients with chronic obstructive pulmonary disease.	10/10 items

## Data Availability

Data sharing is not applicable. No new data were created or analysed in this study.

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
