# Peer review of "Community-Based Interventions in People with Palliative Care Needs: An Integrative Review of Studies from 2017 to 2022"

_healthcare, 2024, doi:10.3390/healthcare12151477_

Round 1

Reviewer 1 Report

Comments and Suggestions for Authors

The authors adress a quite important and sensitive topic that of providing quality of life in people of need palliative care. Although significant initiatives have made huge steps on the matter, this need further investigation and actions.

Comments.

1. Although it's not a systematic review, the authors follow its steps and rules. Regarding the search words used, these are quite poor regarding the descripton of the intervention. The authors should have included words like leisure activities or alternative therapy or spiritual etc.

2. Why the authors chose CASPe checklist for their methodological assessment, especially when they needed to make adaptations? There are other tools for every study design.

3. Table 2 should be re-written on the basis of PIO. For example they don't need to state the whole title of the study, but just the first autor and year. They need to give more precise information on the intervention, the examined outcome and the results. Form the information given, little has to do with the intervention proposed and quality of life. Care plannning as a community intervention was not mentioned. Probably the authors need to clarify what they included under the term of community interventions as well put in section 3.1.5 and 3.1.6.

4. Interdisciplinary home-based teams: an interesting term, but the authors need to further explain what they do and provide.

5. Are all these community based or home based? Which is the most appropriate term? When we say community we describe something that is provided outside the house in a community center, in church or in school. It has its basis of application in the community. In the conclusions the authors used both terms. Should this be included in all sections of the paper : title, methods etc.

6. Avoid presenting same information in tables and text. For example ref 21 cognitive intervention. We need coded presentation of the results (outcome/p value) in the table but in the results we need more information. The same applies for the description of the intervention.

7. Section 4.1 is important as it should present suggestions of implementing the findings. This is not the case. The authors should further enrich this section with information on how this home-community based intervention could be organised and how palliative patients could be referred to it, easily.

Author Response

We would like to thank you for giving us the opportunity to review again and improve our manuscript.

We have considered all suggestions and incorporated them into the revised manuscript, and as a result, we believe our manuscript is stronger. Responses to his comments are written in bold type. We have highlighted in yellow the changes made to the manuscript.

Reviewer 1:

The authors adress a quite important and sensitive topic that of providing quality of life in people of need palliative care. Although significant initiatives have made huge steps on the matter, this need further investigation and actions.

Comments 1: Although it's not a systematic review, the authors follow its steps and rules. Regarding the search words used, these are quite poor regarding the descripton of the intervention. The authors should have included words like leisure activities or alternative therapy or spiritual etc.

Response 1: Thank you very much for your comments, we will take it into account in future reviews. However, in this review, since it is a very specific topic and there are few interventions focused on it, we wanted to do a search with general terms and then sift the search results in order to obtain a greater number of results.

Comments 2: Why the authors chose CASPe checklist for their methodological assessment, especially when they needed to make adaptations? There are other tools for every study design.

Response 2: We have used the CASPe guideline in this systematic review because this guideline ensures a rigorous and structured approach to assessing the quality of studies, identifies potential biases, improves transparency and reproducibility, and facilitates decision-making based on high-quality evidence. This contributes to the reliability and validity of the conclusions of this systematic review and is a guideline used in numerous high-quality systematic reviews.

Comments 3: Table 2 should be re-written on the basis of PIO. For example they don't need to state the whole title of the study, but just the first autor and year. They need to give more precise information on the intervention, the examined outcome and the results. Form the information given, little has to do with the intervention proposed and quality of life. Care plannning as a community intervention was not mentioned. Probably the authors need to clarify what they included under the term of community interventions as well put in section 3.1.5 and 3.1.6.

Response 3: Table 2 has been reviewed and reported as recommended by the reviewer. Thank you very much. Additionally, the term community intervention has been explained at the end of the introduction.

Comments 4: Interdisciplinary home-based teams: an interesting term, but the authors need to further explain what they do and provide.

Response 4: Thank you for your comment, we have enriched the text to improve the expression of concepts.

Comments 5: Are all these community based or home based? Which is the most appropriate term? When we say community we describe something that is provided outside the house in a community center, in church or in school. It has its basis of application in the community. In the conclusions the authors used both terms. Should this be included in all sections of the paper : title, methods etc.

Response 5: Thank you for your comment, as you suggest, community-based services (community includes the home) is more appropriate, so we have changed the term home-based services to community in the text.

Comments 6: Avoid presenting same information in tables and text. For example ref 21 cognitive intervention. We need coded presentation of the results (outcome/p value) in the table but in the results we need more information. The same applies for the description of the intervention.

Response 6: Thank you, we have provided more information about the cognitive results in the results section.

Comments 7: Section 4.1 is important as it should present suggestions of implementing the findings. This is not the case. The authors should further enrich this section with information on how this home-community based intervention could be organised and how palliative patients could be referred to it, easily.

Response 7: Thank you very much for your comments. We have added this information in section 4.2.

Reviewer 2 Report

Comments and Suggestions for Authors

Dear author, your integrative review on community-based interventions for palliative care needs provides valuable insights into this important area of healthcare. The review is well-structured and offers a comprehensive overview of various interventions. However, there are several areas where the manuscript could be strengthened to enhance its clarity, depth, and impact.

Title:

·      Include a timeframe in the title, such as "...Needs: An Integrative Review of Studies from 2017-2022".

·      Fix title space between with and Palliative.

Abstract:

·      In the methodology section, specify the exact date range of the literature search (e.g., "from January 2017 to December 2022"). Say how many databases were searched (five).

·      Mention one or two key findings about these interventions' efficacy in the results.

Introduction:

·      Continue defining palliative care in the first sentence. Include more information about palliative care's holistic approach and goals beyond quality of life.

·      Explain palliative care's global prevalence.

·      Provide a brief overview of community-based palliative care challenges like resource constraints, geographical disparities, and staff shortages. This highlights the significance of your review.

·      Clarify the knowledge gap this review addresses. Why is this review needed because the literature is lacking?

·      The paragraph about Spanish Law 4/2017 is odd. Consider moving this information to a more relevant section or integrating it into the introduction.

·      To improve introduction flow, paragraph transitions could be improved.

Mehods:

·      Study Design: Explain why integrative reviews were chosen. Explain how this method benefits your research question.

·      Strategy for Searching Explain how the search strategy was created using PIO. Describe why these databases were chosen.

·      Criteria for inclusion/exclusion Explain why the 5-year publication limit was chosen. The criteria are clear.

·      Short Data Analysis section. Explain how the data were analyzed to compare intervention outcomes. Describe any statistical methods used.

·      General remarks: Consider adding a section on how ethical issues were addressed, even if only to say this study did not require ethical approval.

Results:

·      To highlight the most significant findings across all intervention types, include a brief summary at the end of the Results section.

Discussion:

·      To improve organization, consider adding subheadings to the discussion, similar to the structure of the Results section. A brief paragraph summarizing the main findings would start the discussion.

·      Consider how the interventions may work together in comprehensive palliative care.

·      Create a limitations section. Discuss study biases, review methodology limitations, and literature gaps.

·      Future Research: In the Conclusions, you mention the need for more research, but in the Discussion, you could provide more detail.

·      You say some interventions are cheap. Discuss intervention cost-effectiveness in greater detail.

·      Telemedicine in the COVID-19 pandemic is intriguing. Discuss how technology may change community-based palliative care.

Conclusion:

·      Address review limitations directly.

·      Promote future research by identifying areas or questions that need more study.

·      Make a strong conclusion about how community-based interventions may improve palliative care quality and accessibility.

Comments on the Quality of English Language

The overall quality of English in this manuscript is good, requiring only minor editing. Specific observations include:

  1. Generally clear and comprehensible writing throughout.
  2. Appropriate use of academic terminology and vocabulary.
  3. Well-structured sentences and paragraphs in most sections.
  4. A few minor issues noted:
    • Spacing error in the title ("withPalliativeCare")
    • Occasional long, complex sentences that could be simplified for clarity
    • Some paragraph transitions could be smoother
    • Minor grammatical errors (e.g., subject-verb agreement) in a few instances

Author Response

We would like to thank you for giving us the opportunity to review again and improve our manuscript.

We have considered all suggestions and incorporated them into the revised manuscript, and as a result, we believe our manuscript is stronger. Responses to his comments are written in bold type. We have highlighted in yellow the changes made to the manuscript.

Reviewer 2:

Dear author, your integrative review on community-based interventions for palliative care needs provides valuable insights into this important area of healthcare. The review is well-structured and offers a comprehensive overview of various interventions. However, there are several areas where the manuscript could be strengthened to enhance its clarity, depth, and impact.

Comments 1:

Title:

  • Include a timeframe in the title, such as "...Needs: An Integrative Review of Studies from 2017-2022".
  • Fix title space between with and Palliative.

Response 1: Thank you for your comment, this information has been added.

Comments 2:

Abstract:

  • In the methodology section, specify the exact date range of the literature search (e.g., "from January 2017 to December 2022"). Say how many databases were searched (five).
  • Mention one or two key findings about these interventions' efficacy in the results.

Response 2: Thank you for your comment, this information has been added.

Comments 3:

Introduction:

  • Continue defining palliative care in the first sentence. Include more information about palliative care's holistic approach and goals beyond quality of life.
  • Explain palliative care's global prevalence.
  • Provide a brief overview of community-based palliative care challenges like resource constraints, geographical disparities, and staff shortages. This highlights the significance of your review.
  • Clarify the knowledge gap this review addresses. Why is this review needed because the literature is lacking?
  • The paragraph about Spanish Law 4/2017 is odd. Consider moving this information to a more relevant section or integrating it into the introduction.
  • To improve introduction flow, paragraph transitions could be improved.

Response 3: Thank you for your comment, this information has been added

Comments 4:

Methods:

  • Study Design: Explain why integrative reviews were chosen. Explain how this method benefits your research question.
  • Strategy for Searching Explain how the search strategy was created using PIO. Describe why these databases were chosen.
  • Criteria for inclusion/exclusion Explain why the 5-year publication limit was chosen. The criteria are clear.
  • Short Data Analysis section. Explain how the data were analyzed to compare intervention outcomes. Describe any statistical methods used.
  • General remarks: Consider adding a section on how ethical issues were addressed, even if only to say this study did not require ethical approval.

Response 4: Thank you for your comment, this information has been added

 Comments 5:

Results:

  • To highlight the most significant findings across all intervention types, include a brief summary at the end of the Results section.

Response 5: Thank you for your comment, this information has been added

Comments 6:

Discussion:

  • To improve organization, consider adding subheadings to the discussion, similar to the structure of the Results section. A brief paragraph summarizing the main findings would start the discussion.
  • Consider how the interventions may work together in comprehensive palliative care.
  • Create a limitations section. Discuss study biases, review methodology limitations, and literature gaps.
  • Future Research: In the Conclusions, you mention the need for more research, but in the Discussion, you could provide more detail.
  • You say some interventions are cheap. Discuss intervention cost-effectiveness in greater detail.
  • Telemedicine in the COVID-19 pandemic is intriguing. Discuss how technology may change community-based palliative care.

Response 6: Thank you for your comment, this information has been added

Comments 7:

Conclusion:

  • Address review limitations directly.
  • Promote future research by identifying areas or questions that need more study.
  • Make a strong conclusion about how community-based interventions may improve palliative care quality and accessibility.

Response 7: Thank you for your comment, the limitations have been included in the discussion section.

Round 2

Reviewer 1 Report

Comments and Suggestions for Authors

The authors have made a significant effort to enrich their manuscript and answer all comments raised.

Yet, I would have to further support my argument regarding the words that were selected for the search. As the authors stated they are too general and this could create different problems. The intervention is just described as community-training and the effect is narrowed to quality of life.

And, regarding the terms community and home based interventions. These are two distinct terms and both used. It would be better for both to be included.

Author Response

We would like to thank you for giving us the opportunity to review again and improve our manuscript.
We have considered all suggestions and incorporated them into the revised manuscript, and as a result, we believe our manuscript is stronger. Responses to his comments are written in bold type. We have highlighted in yellow the changes made to the manuscript.
Reviewer 1:
The authors have made a significant effort to enrich their manuscript and answer all comments raised.
Thank you for your positive comments
Comments 1: Yet, I would have to further support my argument regarding the words that were selected for the search. As the authors stated they are too general and this could create different problems. The intervention is just described as community-training and the effect is narrowed to quality of life. Response 1: Thank you for your comments. We have modified the PIO question and the key words used to clarify the intervention so that it is not so general (line 133-135 and Table 1).

Comments 2: And, regarding the terms community and home based interventions. These are two distinct terms and both used. It would be better for both to be included.
Response 2: Thank you for your comment, we have added to the keywords home care services in the Table 1.
